# The Global Impact of Alcohol Consumption on Premature Mortality and Health in 2016

**DOI:** 10.3390/nu13093145

**Published:** 2021-09-09

**Authors:** Ivneet Sohi, Ari Franklin, Bethany Chrystoja, Ashley Wettlaufer, Jürgen Rehm, Kevin Shield

**Affiliations:** 1Centre for Addiction and Mental Health, Institute for Mental Health Policy Research, 33 Ursula Franklin Street, Toronto, ON M5S 2S1, Canada; Ari.Franklin@camh.ca (A.F.); bethany.chrystoja@ucalgary.ca (B.C.); Ashley.Wettlaufer@camh.ca (A.W.); jtrehm@gmail.com (J.R.); kevin.shield@camh.ca (K.S.); 2Dalla Lana School of Public Health, University of Toronto, 155 College Street, Toronto, ON M5T 3M7, Canada; 3Cumming School of Medicine, University of Calgary, 3330 Hospital Drive NW, Calgary, AB T2N 4N1, Canada; 4Centre for Addiction and Mental Health, Campbell Family Mental Health Research Institute, 33 Russell Street, Toronto, ON M5S 2S1, Canada; 5Department of Psychiatry, University of Toronto, 250 College Street, Toronto, ON M5T 1R8, Canada; 6Institute of Medical Science, University of Toronto, 1 King’s College Circle, Toronto, ON M5S 1A8, Canada; 7Institute of Clinical Psychology and Psychotherapy & Center for Clinical Epidemiology and Longitudinal Studies, Technische Universität Dresden, Chemnitzer Street 46, D-01187 Dresden, Germany; 8Department of International Health Projects, Institute for Leadership and Health Management, I.M. Sechenov First Moscow State Medical University, Trubetskaya Street, 8, b. 2, 119992 Moscow, Russia

**Keywords:** alcohol, burden of disease, death, disability, infectious diseases, non-communicable diseases, injuries, global, policy

## Abstract

This study aimed to estimate the impact of alcohol use on mortality and health among people 69 years of age and younger in 2016. A comparative risk assessment approach was utilized, with population-attributable fractions being estimated by combining alcohol use data from the Global Information System on Alcohol and Health with corresponding relative risk estimates from meta-analyses. The mortality and health data were obtained from the Global Health Observatory. Among people 69 years of age and younger in 2016, 2.0 million deaths and 117.2 million Disability Adjusted Life Years (DALYs) lost were attributable to alcohol consumption, representing 7.1% and 5.5% of all deaths and DALYs lost in that year, respectively. The leading causes of the burden of alcohol-attributable deaths were cirrhosis of the liver (457,000 deaths), road injuries (338,000 deaths), and tuberculosis (190,000 deaths). The numbers of premature deaths per 100,000 people were highest in Eastern Europe (155.8 deaths per 100,000), Central Europe (52.3 deaths per 100,000 people), and Western sub-Saharan Africa (48.7 deaths per 100,000). A large portion of the burden of disease caused by alcohol among people 69 years of age and younger is preventable through the implementation of cost-effective alcohol policies such as increases in taxation.

## 1. Introduction

Alcohol consumption is a leading risk factor for premature mortality and the burden of disease worldwide. Research in developed and developing countries has found that individuals of younger ages are disproportionately affected by alcohol [1,2,3]. For instance, alcohol is estimated to be the leading risk factor for the burden of disease among people 15 to 49 years of age, followed by high body mass index, high blood pressure, and dietary risks [3]. This population also has a large proportion of their expected lifespans remaining, contributes relatively more to the economy, and plays important roles in caring for their families [4].

In response to the burden of disease caused by alcohol, the World Health Organization (WHO), through its Global strategy to reduce the harmful use of alcohol and its Global Action Plan for the prevention and control of NCDs (2013–2020), agreed at the 2010 World Health Assembly to aim for a 10% relative reduction in harmful alcohol use by 2025 [5,6]. Furthermore, the WHO’s Sustainable Development Goals (SDG) 3.4 outlines a targeted one-third reduction by 2030 of premature mortality (i.e., deaths among people 69 years of age and younger) due to noncommunicable diseases, with reductions in alcohol-attributable diseases being key to achieving this goal [7,8]. There is a distinct spectrum of alcohol-attributable diseases and injuries which affect people 69 years of age and younger compared to people 70 years of age and older. Therefore, it is necessary to characterize the disease-specific health impacts of alcohol for the purposes of structuring disease-specific health efforts, for example to inform cancer prevention programs [9].

Given the impact of alcohol consumption on premature mortality, the objective of this study was to estimate the alcohol-attributable mortality and burden of disease globally in 2016, and to examine variations in the alcohol-attributable burden over time, by global burden of disease (GBD) region, age, and sex.

## 2. Materials and Methods

A comparative risk assessment methodology was utilized to estimate the burden of disease attributable to alcohol use in 2016. These estimations were based on the theoretical minimum risk exposure level (TMREL) of lifetime abstention. Lifetime abstention was utilized as a TMREL based on historical precedent, and the observation that lifetime abstainers may have the lowest risk of overall health loss [10]. The population-attributable fraction (PAF) for alcohol use was estimated based on a Levin-based method which combines data on alcohol exposure with corresponding relative risk (RR) estimates [11,12]. Information regarding the methods utilized and the data sources can be found in the Appendix A and in the paper by Shield et al. [1].

### 2.1. Relative Risk Estimates

Alcohol RR estimates for chronic disease outcomes (except from ischemic diseases) were obtained from meta-analyses and based on average drinking (in grams per day) [13]. The lag time between alcohol use and disease occurrence was only modelled for cancer (based on the estimate that there is a 10 year period between exposure and disease outcomes [14]). Heavy episodic drinking (HED) was utilized in the modelling for the RRs for ischemic diseases and injuries [1]. All RR estimates were reviewed and approved by the WHO Technical Advisory Group on Alcohol and Drug Epidemiology. The sources of RR estimates are outlined in Appendix A.

### 2.2. Mortality, Morbidity, and Population Data

Data on mortality, Years of Life Lost (YLL), morbidity measured using Years Lived with Disability (YLD), age, sex, country, year, and by cause of mortality and/or morbidity were obtained from the WHO’s Global Health Estimates [15]. The total burden of disease was measured using Disability Adjusted Life Years (DALYs) lost. Estimates of premature mortality were based on a cut off of deaths which occurred among people 69 years of age and younger [4,16].

Alcoholic cardiomyopathy deaths, YLL, and YLD were estimated using the methods of Manthey and colleagues [17] as they were not directly estimated in the WHO’s Global Health Estimates. The WHO’s road traffic death database [18] was used to determine the fractions of alcohol-attributable motor vehicle deaths which involved a driver and those traffic deaths which involved people other than the driver.

Population data by age, sex, country, and year were obtained from the UN Population Division [19]. Deaths, YLLs, YLDs, and DALYs lost were aggregated into five-year age groups, beginning at 0 years until 84 years, followed by the category of 85 years and older; alcohol PAFs were applied to these age groupings.

Data were aggregated by GBD region (see: http://ghdx.healthdata.org (accessed on 1 January 2021) for regional groupings) and by Human Development Index (HDI) region (see Appendix A for HDI groupings). HDI categories were obtained from the United Nations Development Programme [20]. The HDI is based on having a long and healthy life (i.e., life expectancy at birth), being knowledgeable (i.e., expected years of schooling and mean years of schooling for adults 25 years of age and older), and having a decent standard of living (i.e., Gross National Income *per capita*) [20].

The 95% uncertainty intervals (see Appendix A) were based on a set of 1000 simulations of all lowest level parameters (i.e., parameters sampled from their respective error distributions). These parameters were then used to estimate 1000 simulated estimates of the alcohol-attributable burden of disease. From these simulations, the 2.5th and 97.5th percentiles were utilized for the 95% uncertainty intervals.

Analyses were performed using the statistical software package R [21].

## 3. Results

In 2016, there were 2.0 million premature deaths and 117.2 million DALYs lost globally due to alcohol use, representing 7.1% of all premature deaths and 5.5% of all DALYs lost in that year (see Table 1 and Table 2). In contrast, 3.2% of all deaths and 3.0% of all DALYs lost among people 70 years of age and older were attributable to alcohol consumption. An estimated 70.7% of all alcohol-attributable deaths and 89.2% of all alcohol-attributable DALYs lost globally in 2016 were premature, i.e., among those 69 years of age and younger. In comparison, 52.0% of all deaths and 81.8% of all DALYs lost globally in 2016 were premature. The alcohol-attributable deaths and DALYs lost among those 69 years of age and younger were greater among men (1.6 million deaths and 90.9 million DALYs lost) compared to women (0.5 million deaths and 26.3 million DALYs lost). The largest proportion of premature deaths that were attributable to alcohol occurred among people 30–39 years of age (13.3%) and 20–29 years of age (13.0%). See Appendix A for data on sex-specific alcohol-attributable deaths, YLL, YLD, and DALYs lost.

The leading causes of the burden of premature alcohol-attributable deaths were cirrhosis of the liver (457,000 deaths), road injuries (338,000 deaths), and tuberculosis (190,000 deaths) (see Figure 1). Road injuries and cirrhosis of the liver were the leading causes of alcohol-attributable deaths among those aged 0 to 39 and 40 to 69 years of age, respectively. The proportion of alcohol-attributable deaths due to road injuries decreased with age from 100% of all alcohol-attributable deaths among those aged 0–14 years to 7.0% among those aged 60–69 years. The proportion of alcohol-attributable deaths due to cirrhosis of the liver increased with age, peaking at 26.4% of all alcohol-attributable deaths among those 50–59 years of age. The proportion of alcohol-attributable deaths due to tuberculosis increased with age, peaking at 10.7% of all alcohol-attributable deaths among those 40–49 years of age.

### 3.1. Alcohol-Attributable Burden of Disease by Region

The numbers of premature alcohol-attributable deaths and DALYs lost per 100,000 people showed large variations globally (see Figure 2 and Figure 3). The numbers of premature alcohol-attributable deaths were highest in Eastern Europe (155.8 deaths per 100,000), Central Europe (52.3 deaths per 100,000 people), and Western sub-Saharan Africa (48.7 deaths per 100,000). In 2016, the two leading contributors to alcohol-attributable deaths among all regions were either cirrhosis of the liver or road injuries, except for three regions: Asia Pacific with self-harm, Southern sub-Saharan Africa with tuberculosis, and Eastern Europe with ischaemic heart disease being the largest contributors, respectively (see Figure 3). The second largest contributors to alcohol-attributable deaths in Southern sub-Saharan Africa and Eastern Europe were HIV/AIDS and alcohol use disorders, respectively. Appendix A outlines the burden of alcohol-attributable premature YLL and YLD globally. Appendix A outlines the burden of alcohol-attributable premature YLL and YLD by GBD region.

### 3.2. Alcohol-Attributable Burden of Disease by Human Development Index

The burden of premature alcohol-attributable deaths and DALYs lost varied by HDI region (see Figure 4). The number of alcohol-attributable deaths was highest in countries with a very high HDI (43.3 deaths and 2339.9 DALYs lost per 100,000), followed by low HDI countries (33.7 deaths and 1966.1 DALYs lost per 100,000), high HDI countries (26.8 deaths and 1502.3 DALYs lost per 100,000) and medium HDI countries (24.8 deaths and 1392.5 DALYs lost per 100,000). The leading cause of death was cirrhosis of the liver in very high HDI countries (8.2 deaths per 100,000), low HDI countries (7.4 deaths per 100,000), and medium HDI countries (6.6 deaths per 1,000,000), and was road injuries in high HDI countries (5.9 deaths per 100,000). For DALYs lost, the leading contributor to the premature alcohol-attributable burden of disease was alcohol use disorders for very-high HDI countries (449.3 DALYs lost per 100,000), liver cirrhosis for medium HDI countries (297.3 DALYs lost per 100,000 people), and road injuries for low HDI countries (431.2 DALYs lost per 100,000 people) and high HDI countries (354.4 DALYs lost per 100,000). Appendix A outlines the burden of premature YLL and YLD by HDI region.

## 4. Discussion

The results of this study indicate that alcohol-attributable deaths and health loss occurred among people relatively young in age. The proportions of alcohol-attributable deaths and DALYs lost that were premature were greater than the proportions of all-cause deaths and DALYs lost that were considered premature. This indicates that alcohol use disproportionately affects the health of people who are younger in age. The cause composition of the premature alcohol-attributable burden is unique when compared to the burden among people 70 years of age and older, with cirrhosis of the liver, road injuries, and tuberculosis being the primary contributors to this burden. Furthermore, regional and societal development-based variations in the magnitude of the premature alcohol-attributable burden of disease and the cause composition of this burden were observed.

This study modelled both the detrimental and protective effects of alcohol consumption on health. Specifically, alcohol consumed at low amounts, and not on HED occasions has a protective effect on diabetes, ischemic heart disease, and ischemic stroke [13]. This study found that for the premature disease burden, the detrimental effects of alcohol at the population level outweighed the protective effects. At the individual level, the net effect of alcohol consumption on overall health is unknown; however, a recent modelling study found no level of alcohol consumption that provided a net health benefit [10].

The premature burden of disease attributable to alcohol consumption was characterized by tuberculosis, liver cirrhosis, and injuries. Liver cirrhosis is mainly linked to the overall volume of drinking, while injuries attributable to alcohol are mostly related to intoxication (i.e., binge alcohol consumption) [13]. Tuberculosis risk (i.e., the impact of alcohol on the immune system) is related both to the overall volume of alcohol consumed and binge drinking; however, due to the lack of studies, the impact of alcohol use on tuberculosis was modelled based only on the overall volume of alcohol consumed [13]. Therefore, both overall volume of alcohol consumed and drinking to intoxication are factors leading to the premature burden of disease attributable to alcohol consumption.

Tuberculosis remains an enormous public health concern globally, especially in low and medium HDI countries [22]. The treatment of tuberculosis and the interaction between HIV/AIDS and tuberculosis are key public health priorities [23]. Alcohol use is a key risk factor for both diseases (Morojele et al., this issue), which if addressed can substantially reduce the health burden of tuberculosis and HIV/AIDS. The burden of disease due to liver cirrhosis was high in all HDI categories and in most GBD regions. The burden of alcoholic liver cirrhosis is affected by multiple risk factors which interact with alcohol, including hepatitis B and C infections, obesity, and socio-economic status [24]. The burden of alcohol-related injuries is problematic as investment in preventing mortality from injury has fallen behind other causes of death, such as HIV/AIDS and reproductive health [21]. Furthermore, mental health concerns have been overlooked in terms of public health programming, especially in young people [10] where injuries and neuropsychiatric conditions are greatly impacted by alcohol consumption [22].

The burden of premature disease attributable to alcohol consumption was highest in Eastern Europe, Central Europe, and Western sub-Saharan Africa. The Central and Eastern Europe region have a high overall volume of alcohol consumption and a high prevalence of HED [5,6]. Alcohol control policy measures, including increases in alcohol prices and decreases in availability, have been implemented in the Eastern Europe region and have resulted in marked downward shifts in mortality and the burden of disease [25]. The Western sub-Saharan Africa region has a relatively low overall volume of alcohol consumption. The burden of alcohol-attributable premature disease in this region was driven mainly by infectious diseases, liver cirrhosis, and injuries. Cirrhosis-related deaths doubled in the sub-Saharan Africa region between 1980 and 2010, with hepatitis B virus, hepatitis C virus, and alcohol use being contributing factors to this increase [26]. Furthermore, treatment of liver cirrhosis is unavailable in most parts of sub-Saharan Africa, due to a shortage of hepatologists and gastroenterologists, interventional radiologists, hepatobiliary surgeons, and pathologists [27].

### 4.1. Limitations

The methods used in this paper are limited by several factors. Firstly, estimates of alcohol consumption came from surveys which are susceptible to numerous biases which lead to an underestimation of alcohol use. Per capita consumption of alcohol is utilized to estimate the volume of alcohol use among drinkers to avoid bias; however, no correction exists for the prevalence of HED. This study did not fully account for the interaction between alcohol use and other risk factors, such as smoking (increased risk of cancer [28]), hepatitis B and C (increased risk of liver cirrhosis [29]), and obesity (increased risk of liver cirrhosis [24,30]). Furthermore, the study did not account for the differential alcohol RRs by socio-economic status. Furthermore, although depression has been shown to be causally related to alcohol consumption, it was not included in the estimates of the alcohol-attributable burden of disease due to depression also causally increasing alcohol consumption [13].

This study is also limited as deaths and health loss due to interpersonal harm are based on the alcohol consumption of the person who experiences the harm and not the alcohol consumption of the person who inflicts the harm. This is due to the relative risks of injuries from assault being based on the person who experiences the harm and not the person who is inflicting the harm [13]. Therefore, estimates of intentional harm are likely underestimated for children and women who are often victims of alcohol-related violence [31]. It is important to note that violence against women and children is a major public health, social policy, and human rights concern that spans disciplines and geographical boundaries [32,33,34,35]. Globally, domestic violence is one of the largest sources of non-fatal injuries to women and children [36], resulting in avoidable inequities in health status, and increases in the risk of mental health and physical conditions [37].

### 4.2. Health Policies

The health harms and inequities outlined in the paper should be considered in the context of population-level interventions which can reduce the alcohol-attributable burden of disease and are sustainable, scalable, and politically, economically, and technically feasible [38]. Several alcohol interventions have been designated as “best buys” by the WHO as they are more cost-effective than most other interventions designed for other risk factors [39]. These include increases in taxation and restrictions on availability and marketing. Other policies include WHO cost-effective “very good buys,” such as enactment and enforcement of impaired-driving laws and blood-alcohol-concentration limits [39,40].

The need to intervene to reduce the burden of premature health loss attributable to alcohol consumption can be viewed under the framework of utilitarian ageism. The framework of utilitarian ageism, which is often observed in medical practice, states that there should be prioritization of treatments and interventions for health loss among the young (as the old have lived longer) [41,42].

Despite the majority of the burden of disease attributable to alcohol consumption occurring among people 0 to 69 years of age, alcohol leads to a substantial burden of disease among people 70 years of age and older. Interventions such as the WHO best buys and very best buys should also be prioritized to reduce the burden among people 70 years of age and older. Furthermore, these policies should apply equally to beer, wine, spirits, and other alcoholic beverages as the harm caused by alcohol is based on ethanol content regardless of whether the ethanol is consumed in the form of beer, wine, or spirits (with the exception of alcohol poisonings which are caused predominately by the consumption of spirits) [13].

## 5. Conclusions

Alcohol consumption remains a leading risk factor for the burden of disease, especially among people younger in age. Given the high global alcohol-attributable burden of disease, the development and implementation of cost-effective alcohol control policies can further reduce in the near future the social, economic, and health burdens resulting from the use of alcohol.

## Figures and Tables

**Figure 1 nutrients-13-03145-f001:**
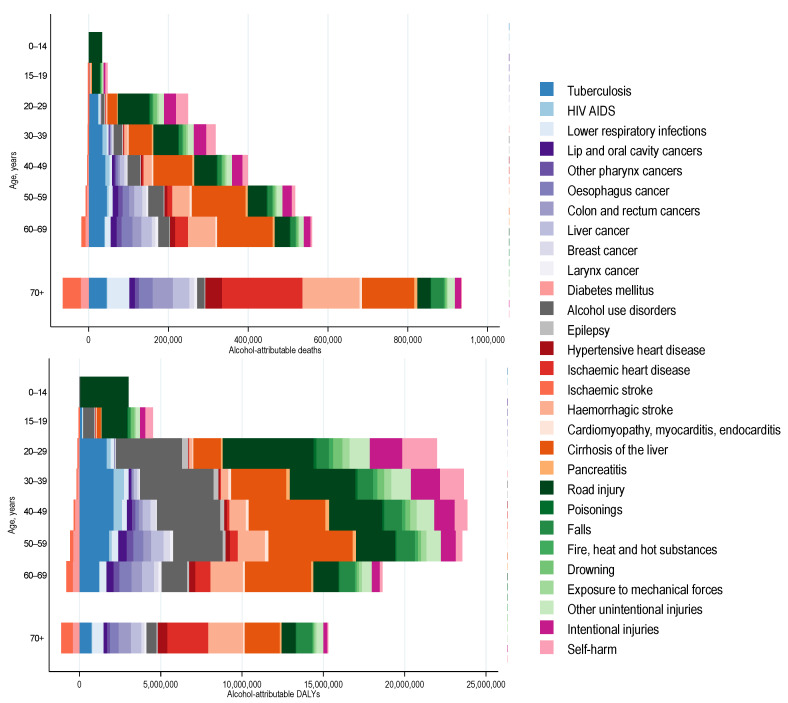
Alcohol-attributable deaths and disability adjusted life years (DALYs) lost by age.

**Figure 2 nutrients-13-03145-f002:**
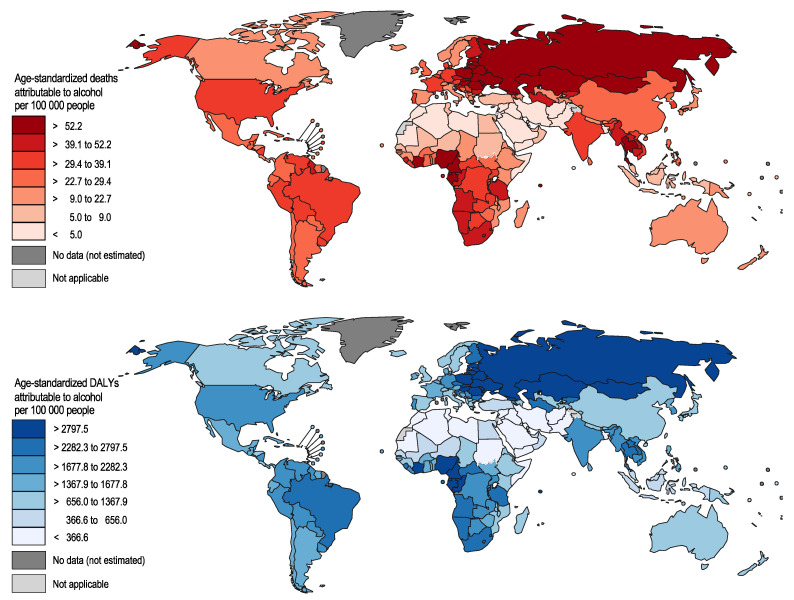
Alcohol-attributable deaths and disability adjusted years of life lost globally in 2016 among people 0 to 69 years of age.

**Figure 3 nutrients-13-03145-f003:**
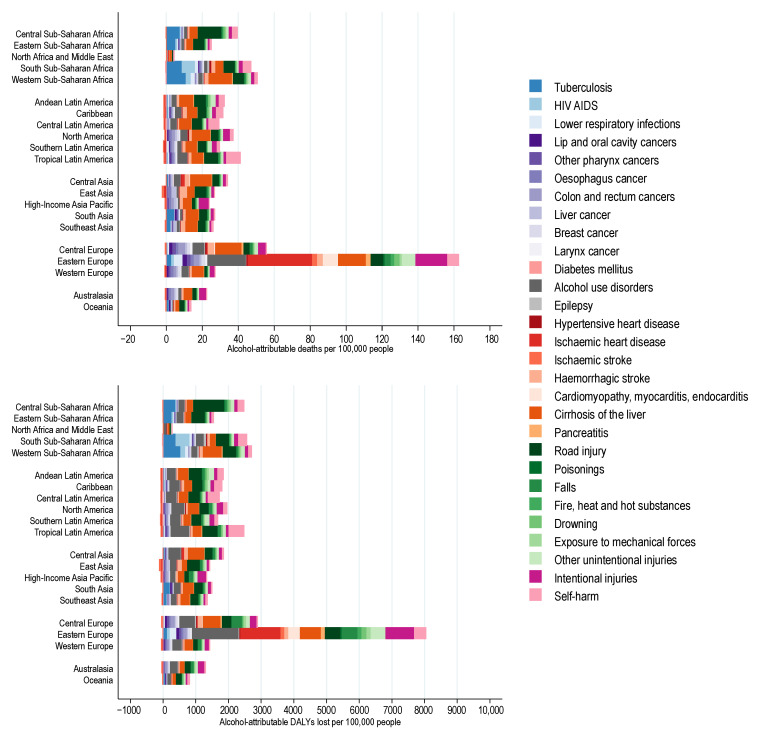
Alcohol-attributable deaths and disability adjusted life years (DALYs) lost among people 0 to 70 years of age by global burden of disease region.

**Figure 4 nutrients-13-03145-f004:**
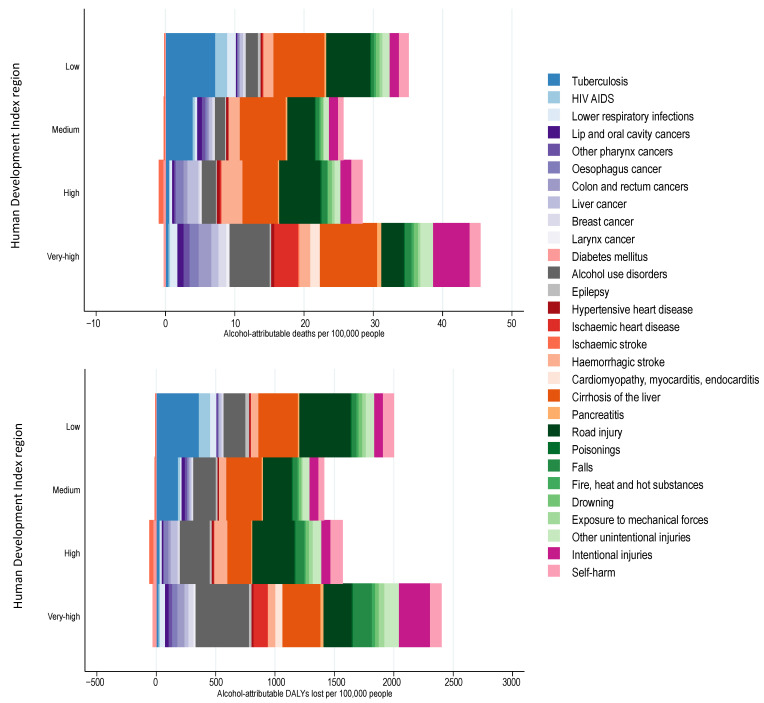
Alcohol-attributable deaths and disability adjusted life years (DALYs) lost among people 0 to 70 years of age by human development index region.

**Table 1 nutrients-13-03145-t001:** Alcohol-attributable deaths globally in 2016, by cause and age.

Cause of Disease or Injury	Alcohol-Attributable Deaths								Population Attributable Fraction (%)							
	0 to 14	15 to 19	20 to 29	30 to 39	40 to 49	50 to 59	60 to 69	≥70	0 to 14	15 to 19	20 to 29	30 to 39	40 to 49	50 to 59	60 to 69	≥70
All Causes	33,939	47,719	248,762	317,457	396,552	510,448	542,012	870,937	0.5	6.9	13.0	13.3	12.0	9.2	5.9	3.2
Communicable, maternal, perinatal and nutritional conditions	0	3073	30,471	50,497	58,869	60,646	55,847	102,481	0.0	1.5	5.4	6.8	8.5	9.1	6.5	4.0
Tuberculosis	0	2299	23,738	34,975	42,291	46,097	40,121	46,762	0.0	8.5	22.0	24.7	25.4	22.0	17.5	12.7
HIV AIDS	0	269	3444	10,566	10,220	4352	1284	302	0.0	1.0	3.0	3.5	3.8	3.5	3.1	2.3
Lower respiratory infections	0	505	3289	4956	6358	10,197	14,442	55,417	0.0	2.2	6.2	8.4	8.5	7.4	5.1	4.1
Noncommunicable diseases	0	5997	42,304	111,176	203,371	330,932	392,643	656,706	0.0	3.2	7.7	11.2	9.8	7.6	5.0	2.8
Malignant neoplasms	0	0	1094	9183	30,653	73,936	102,619	150,215	0.0	0.0	0.9	3.4	4.5	5.2	4.8	3.6
Lip and oral cavity cancer	0	0	224	2542	6678	13,732	14,915	14,085	0.0	0.0	7.9	30.2	32.2	35.4	33.7	27.8
Other pharynx cancers	0	0	63	975	4250	11,026	12,356	9922	0.0	0.0	7.3	28.5	34.6	39.9	38.1	29.6
Oesophagus cancer	0	0	53	712	4519	16,516	27,076	34,070	0.0	0.0	3.5	12.8	18.6	22.4	21.7	17.2
Colon and rectum cancers	0	0	248	1754	4823	12,582	22,287	50,866	0.0	0.0	3.9	9.6	10.8	12.4	12.8	11.4
Liver cancer	0	0	507	3199	10,383	20,080	25,985	41,273	0.0	0.0	5.6	11.5	11.8	12.4	12.6	12.4
Breast cancer	0	0	178	3003	6743	10,445	9238	12,415	0.0	0.0	2.4	7.1	7.4	8.1	7.4	6.5
Larynx cancer	0	0	13	210	1418	4753	6849	7283	0.0	0.0	4.2	17.9	21.7	24.6	24.0	20.3
Diabetes mellitus	0	−41	−362	−681	−1958	−4474	−7816	−19,753	0.0	−1.1	−2.8	−2.7	−2.6	−2.3	−2.0	−2.2
Alcohol use disorders	0	967	7748	21,406	31,222	38,315	28,103	17,804	100.0	100.0	100.0	100.0	100.0	100.0	100.0	100.0
Epilepsy	0	694	3278	3019	2651	2177	1844	2751	0.0	5.9	15.2	17.3	18.7	16.9	14.4	11.7
Cardiovascular diseases	0	680	4328	13,070	30,099	65,934	107,708	347,811	0.0	1.8	3.2	4.2	4.1	3.8	3.1	3.0
Hypertensive heart disease	0	40	457	1260	3204	7079	12,846	41,571	0.0	3.1	9.1	10.2	10.4	9.4	8.2	6.8
Ischaemic heart disease	0	85	139	1195	3027	12,654	32,002	201,657	0.0	0.8	0.3	0.8	0.8	1.3	1.8	3.3
Ischaemic stroke	0	−11	−75	−174	−838	−2823	−10,280	−45,068	0.0	−0.7	−1.3	−1.5	−2.2	−2.1	−2.0	−2.1
Haemorrhagic stroke	0	476	3044	7768	20,968	43,205	68,579	142,988	0.0	5.7	10.4	11.3	11.8	10.8	9.7	9.2
Cardiomyopathy, myocarditis, endocarditis	0	89	763	3021	3738	5819	4562	6662	0.0	1.9	5.0	12.3	11.5	12.3	7.5	3.8
Digestive diseases	0	3697	26,027	61,967	102,541	139,846	144,097	138,180	0.0	12.8	28.4	37.8	39.9	35.6	27.9	13.9
Cirrhosis of the liver	0	3593	24,369	58,145	97,529	134,702	139,105	130,695	0.0	28.8	49.6	54.0	55.3	52.5	47.8	38.9
Pancreatitis	0	104	1658	3822	5012	5144	4993	7485	0.0	11.0	27.3	32.2	31.6	28.3	23.2	18.5
Injuries	33,939	38,649	175,987	155,784	134,313	118,870	93,522	111,750	5.3	12.8	21.9	24.0	25.2	23.2	18.4	12.1
Unintentional injuries	33,939	28,303	116,260	101,421	94,461	87,006	73,167	94,404	5.9	16.0	26.1	27.3	27.6	24.6	19.1	12.0
Road injury	33,939	20,533	78,740	62,508	56,129	48,048	37,797	33,073	23.1	20.2	29.1	30.0	30.0	27.5	24.7	20.8
Poisonings	0	409	2543	2273	2342	1969	1718	1480	0.0	10.0	20.7	22.9	21.8	19.3	13.9	9.5
Falls	0	809	5398	7347	9340	11,524	12,814	32,138	0.0	10.3	22.4	24.6	25.4	20.9	13.5	8.8
Fire, heat and hot substances	0	531	2813	3106	2614	2773	2362	2960	0.0	6.7	14.7	17.5	20.9	20.1	15.5	10.4
Drowning	0	2708	8553	7214	5983	5109	4223	4320	0.0	11.6	23.4	26.0	26.1	22.6	17.3	11.5
Exposure to mechanical forces	0	786	4655	4573	4114	3443	2311	1635	0.0	10.8	23.0	24.9	25.8	23.1	17.7	10.5
Other unintentional injuries	0	2526	13,558	14,400	13,940	14,141	11,942	18,798	0.0	10.4	22.0	24.4	25.1	22.9	16.8	11.3
Intentional injuries	0	10,346	59,726	54,363	39,851	31,863	20,355	17,346	0.0	8.2	16.6	19.7	20.8	20.0	16.2	12.3
Self-harm	0	4491	29,477	31,085	26,207	23,767	16,508	15,484	0.0	8.5	18.4	23.0	23.8	21.9	17.1	12.8
Interpersonal violence	0	5855	30,249	23,278	13,645	8096	3847	1862	0.0	11.1	20.7	22.4	22.0	20.1	16.6	11.1

**Table 2 nutrients-13-03145-t002:** Alcohol-attributable disability adjusted life years lost globally in 2016, by cause for people 0 to 69 years of age.

Cause of Disease or Injury	Alcohol-Attributable DALYs (100,000 s)								Population Attributable Fraction (%)							
	0 to 14	15 to 19	20 to 29	30 to 39	40 to 49	50 to 59	60 to 69	≥70	0 to 14	15 to 19	20 to 29	30 to 39	40 to 49	50 to 59	60 to 69	≥70
All Causes	302.5	451.5	2189.8	2348.6	2349.6	2297.1	1783.2	1419.4	0.5	5.0	9.8	9.9	9.0	7.4	5.1	3.0
Communicable, maternal, perinatal and nutritional conditions	0.0	24.4	215.6	302.8	293.1	238.9	167.2	148.9	0.0	1.1	4.0	5.4	6.7	7.4	5.9	3.9
Tuberculosis	0.0	18.5	168.5	210.4	210.8	182.6	122.0	78.6	0.0	8.5	22.0	24.6	25.4	22.1	18.0	13.0
HIV AIDS	0.0	2.1	24.9	64.0	52.1	18.1	4.2	0.7	0.0	1.0	3.0	3.5	3.8	3.5	3.1	2.2
Lower respiratory infections	0.0	3.8	22.1	28.5	30.2	38.2	41.0	69.6	0.0	2.2	6.2	8.4	8.5	7.4	5.3	4.0
Noncommunicable diseases	6.3	111.5	657.4	975.5	1208.1	1405.3	1191.1	982.8	0.0	2.6	6.0	7.2	6.6	5.6	4.0	2.4
Malignant neoplasms	0.0	0.0	8.4	70.9	184.2	337.5	338.7	263.9	0.0	0.0	1.1	4.6	5.7	6.3	5.5	4.2
Lip and oral cavity cancer	0.0	0.0	1.5	14.5	31.8	52.2	43.1	22.8	0.0	0.0	7.7	30.0	32.2	35.4	33.8	28.7
Other pharynx cancers	0.0	0.0	0.4	5.5	20.0	41.6	35.4	16.6	0.0	0.0	7.1	28.4	34.6	39.9	38.2	30.9
Oesophagus cancer	0.0	0.0	0.3	4.0	21.0	61.6	76.1	53.9	0.0	0.0	3.4	12.8	18.5	22.4	21.7	17.8
Colon and rectum cancers	0.0	0.0	1.6	10.1	23.0	47.9	63.9	75.8	0.0	0.0	3.8	9.6	10.8	12.4	12.8	11.7
Liver cancer	0.0	0.0	3.3	18.2	48.8	75.4	73.3	63.5	0.0	0.0	5.5	11.5	11.8	12.4	12.6	12.4
Breast cancer	0.0	0.0	1.2	17.4	32.9	40.8	27.3	19.2	0.0	0.0	2.4	7.1	7.5	8.2	7.4	6.7
Larynx cancer	0.0	0.0	0.1	1.2	6.7	18.1	19.7	12.0	0.0	0.0	4.1	17.9	21.7	24.6	24.1	20.8
Diabetes mellitus	0.0	−0.7	−7.4	−13.2	−27.2	−37.3	−40.7	−40.2	0.0	−1.3	−3.1	−3.0	−3.1	−2.6	−2.4	−2.2
Alcohol use disorders	6.3	68.0	407.7	451.8	389.3	304.5	157.4	60.6	100.0	100.0	100.0	100.0	100.0	100.0	100.0	100.0
Epilepsy	0.0	9.2	39.3	30.9	24.5	17.7	12.1	9.5	0.0	5.8	14.4	16.0	17.1	15.6	13.9	11.4
Cardiovascular diseases	0.0	5.2	29.2	75.1	142.0	244.7	303.4	462.0	0.0	1.6	2.8	3.9	3.8	3.6	2.9	2.8
Hypertensive heart disease	0.0	0.3	3.0	7.5	16.2	28.1	38.4	58.5	0.0	3.1	9.1	10.0	10.2	9.3	8.3	6.8
Ischaemic heart disease	0.0	0.7	1.1	7.1	14.7	47.5	91.8	252.0	0.0	0.8	0.3	0.8	0.7	1.3	1.8	3.1
Ischaemic stroke	0.0	−0.2	−1.2	−2.4	−8.3	−19.5	−40.0	−71.5	0.0	−1.1	−2.0	−2.3	−3.1	−2.8	−2.3	−2.2
Haemorrhagic stroke	0.0	3.7	21.2	45.5	101.5	166.5	199.6	213.7	0.0	5.8	10.5	11.3	11.9	10.8	9.9	9.0
Cardiomyopathy, myocarditis, endocarditis	0.0	0.7	5.1	17.4	17.9	22.1	13.4	9.3	0.0	1.8	4.7	11.9	11.3	11.9	7.5	3.8
Digestive diseases	0.0	29.8	180.2	360.1	495.3	538.1	420.2	227.0	0.0	12.4	26.6	34.7	36.1	33.2	26.6	14.6
Cirrhosis of the liver	0.0	29.0	169.1	338.1	471.3	518.6	405.8	215.7	0.0	28.9	49.5	53.9	55.3	52.5	48.3	39.5
Pancreatitis	0.0	0.8	11.1	22.0	23.9	19.5	14.4	11.3	0.0	11.1	27.2	32.1	31.5	28.2	23.5	18.8
Injuries	296.2	315.7	1316.8	1070.3	848.4	652.9	425.0	287.7	4.9	12.7	21.8	23.7	24.8	22.8	19.0	13.2
Unintentional injuries	296.2	236.1	904.5	744.6	645.8	522.1	359.6	256.8	5.6	15.6	25.6	26.5	26.8	24.0	19.7	13.4
Road injury	296.2	158.6	557.9	403.0	325.7	241.0	153.7	88.0	23.1	20.3	29.1	30.0	30.2	28.0	25.8	22.4
Poisonings	0.0	3.3	18.2	14.1	12.2	8.2	5.5	2.6	0.0	10.2	20.7	22.9	21.9	19.5	14.6	9.8
Falls	0.0	14.4	76.6	95.2	109.1	113.2	95.6	97.9	0.0	11.9	23.0	24.6	25.2	21.8	16.4	10.7
Fire, heat and hot substances	0.0	5.5	25.9	26.7	21.5	18.1	11.7	7.4	0.0	7.6	15.8	18.4	21.2	20.0	16.7	11.6
Drowning	0.0	20.3	58.0	42.0	29.1	19.8	12.6	7.1	0.0	11.6	23.3	25.9	26.0	22.4	17.6	11.6
Exposure to mechanical forces	0.0	8.1	43.6	43.0	40.1	32.2	20.2	11.0	0.0	11.5	23.4	25.1	26.0	23.6	19.8	13.9
Other unintentional injuries	0.0	25.9	124.3	120.5	108.2	89.6	60.4	42.9	0.0	10.2	21.0	22.5	23.0	20.7	16.5	11.3
Intentional injuries	0.0	79.5	412.3	325.7	202.7	130.8	65.3	30.9	0.0	8.2	16.4	19.1	20.0	19.0	16.0	12.0
Self-harm	0.0	33.7	198.1	180.0	125.9	90.9	48.6	24.7	0.0	8.5	18.4	23.0	23.7	21.9	17.5	12.8
Interpersonal violence	0.0	45.9	214.2	145.7	76.8	39.9	16.7	6.2	0.0	10.9	20.5	22.1	21.8	20.2	17.3	12.0

## Data Availability

All statistical code (i.e., R code) and input files used to produce the results presented in this paper are available to the general public. To obtain the code and input files, please contact the author, Kevin Shield (Kevin.Shield@camh.ca).

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
