# Peer review of "The Global Impact of Alcohol Consumption on Premature Mortality and Health in 2016"

_nutrients, 2021, doi:10.3390/nu13093145_

Round 1
Reviewer 1 Report
In this study, the authors estimated the effect on mortality and health attributable to alcohol use in people younger than 70 years in 2016. The topic is very interesting since alcohol consumption is the leading risk factor for premature mortality among young people. The mortality and health information were obtained from the Global Health Observatory. They found that, in 2016, in subjects 69 years of age or younger, the 7.1% and 5.5% of all deaths and DALYs lost in that year were attributable to alcohol. The leading causes of alcohol attributable deaths were liver cirrhosis, road injuries and tubercolosis. The largest part of premature deaths attributable to alcohol use occurred among people younger than 40 years of age and in people of Eastern Europe. The number of alcohol attributable deaths was highest in countries at very high Human Development Index (HDI) and low HDI.
This study has some limitations: the estimation of alcohol consumption was derived by surveys; the interaction with concomitant diseases (obesity, diabetes mellitus, and others) are missing. However this points have been discussed in the text.
The authors could discuss the mechanisms by which alcohol could increase mortality in young people
Author Response
In this study, the authors estimated the effect on mortality and health attributable to alcohol use in people younger than 70 years in 2016. The topic is very interesting since alcohol consumption is the leading risk factor for premature mortality among young people. The mortality and health information were obtained from the Global Health Observatory. They found that, in 2016, in subjects 69 years of age or younger, the 7.1% and 5.5% of all deaths and DALYs lost in that year were attributable to alcohol. The leading causes of alcohol attributable deaths were liver cirrhosis, road injuries and tubercolosis. The largest part of premature deaths attributable to alcohol use occurred among people younger than 40 years of age and in people of Eastern Europe. The number of alcohol attributable deaths was highest in countries at very high Human Development Index (HDI) and low HDI.
This study has some limitations: the estimation of alcohol consumption was derived by surveys; the interaction with concomitant diseases (obesity, diabetes mellitus, and others) are missing. However this points have been discussed in the text.
The authors could discuss the mechanisms by which alcohol could increase mortality in young people.
Thank you for the comments
We have amended the paper to so that the mechanisms by which alcohol could increase mortality in young people are now discussed.
Reviewer 2 Report
The topic is important and relevant for policy makers. The used methods are well chosen, however some further insights might enhance the easiness to read.
Recently several studies have been published showing positive effects of specific, light/moderate alcohol consumption eg wine. Thus, the discussion section should also address these findings putting them into the 'global' context.
Author Response
Thank you for the comments on the paper. We have responded in bold below.
The topic is important and relevant for policy makers. The used methods are well chosen, however some further insights might enhance the easiness to read.
Recently several studies have been published showing positive effects of specific, light/moderate alcohol consumption eg wine.
We have amended the paper so that we now discuss studies showing positive effects of alcohol on diabetes and ischemic diseases (i.e., heart disease and stroke).
We have also amended the manuscript to indicate that the harms caused by alcohol the harm caused by alcohol is based on the ethanol content regardless of if the ethanol is consumed in the form of beer, wine or spirits (with the exception of alcohol poisonings which are caused predominately by the consumption of spirits).
The potential biological mechanism for wine being beneficial on health is due to the presence of antioxidants in particular resveratrol; however, the benefit from the resveratrol is likely very minimal (see: Lachenmeier, D. W., Godelmann, R., Witt, B., Riedel, K., & Rehm, J. (2014). Can resveratrol in wine protect against the carcinogenicity of ethanol? A probabilistic dose‐response assessment. International journal of cancer, 134(1), 144-153.) Overall, the effects of wine on health are likely due to consuming alcohol with meals, decreased risk of binge drinking and confounding.
Thus, the discussion section should also address these findings putting them into the 'global' context.
We have revised the discussion to detail the global context of the burden of premature disease attributable to alcohol consumption.